# Superlattices, Bonding-Antibonding, Fermi Surface Nesting, and Superconductivity

Jose A. Alarco [1,2,3,*] and Ian D. R. Mackinnon [3,4]

1   School of Chemistry and Physics, Faculty of Science, Queensland University of Technology, Brisbane, QLD 4000, Australia
2   Centre for Materials Science, Queensland University of Technology, Brisbane, QLD 4000, Australia
3   Centre for Clean Energy Technologies and Practices, Queensland University of Technology, Brisbane, QLD 4000, Australia; ian.mackinnon@qut.edu.au
4   School of Earth and Atmospheric Sciences, Faculty of Science, Queensland University of Technology, Brisbane, QLD 4000, Australia
*   Correspondence: jose.alarco@qut.edu.au

**Abstract:** Raman and synchrotron THz absorption spectral measurements on $MgB_2$ provide experimental evidence for electron orbital superlattices. In earlier work, we have detected THz spectra that show superlattice absorption peaks with low wavenumbers, for which spectral density evolves and intensifies after cooling below the superconducting transition temperature for $MgB_2$. In this work, we show how these observations indicate a direct connection to superconducting properties and mechanisms. Bonding–antibonding orbital character is identified in calculated electronic band structures and Fermi surfaces consistent with superlattice structures along the *c*-axis. DFT calculations show that superlattice folding of reciprocal space generates Brillouin zone boundary reflections, Umklapp processes, and substantially enhances nesting relationships. Tight binding equations are compared with expected charge density waves from nesting relationships and adjusted to explicitly accommodate these linked processes. Systematic analysis of electronic band structures and Fermi surfaces allows for direct identification of Cooper pairing and the superconducting gap, particularly when the k-grid resolution of a calculation is suitably calibrated to structural parameters. Thus, we detail a robust and accurate DFT re-interpretation of BCS superconductivity for $MgB_2$.

**Keywords:** density functional theory; superlattice; electronic band structure; orbital; bonding; antibonding; Fermi surface; nesting; charge density wave; superconductivity; mechanisms; $MgB_2$

## 1. Introduction

Density functional theory (DFT) calculations have contributed, and continue to contribute, substantially to our improved understanding of the properties of materials [1–8], including those of superconductors [9–11]. For example, using DFT, binary metal hydrides were predicted to be superconductors at or near room temperature, although requiring very high pressures (>100 GPa) to form as stable phases [12–23]. Several predicted materials have been successfully synthesized and shown to undergo a superconducting transition close to the predicted temperatures [24–28]. Predicted superconducting transition temperatures ($T_c$) are conventionally estimated via calculation of the electron–phonon coupling (EPC), an important and common practice for DFT calculations on superconductors [29,30]. This practice follows learnings from the Eliashberg or Migdal–Eliashberg theory, often making use of the simplified McMillan equations for determining $T_c$ [31–33].

An EPC calculation integrates the product of the density of electronic states and phonon states in all directions to obtain a weighted average value. However, density of states approaches typically result in a compromise with fine granular detail, particularly when precise geometric information from detailed dispersion relationships is lacking [34].

Full phonon dispersions and electronic band structures include increased detail of geometric information; however, while this detail is available, it is seldom extracted in full or utilised. For compounds with more complex compositions and crystal structures than metal hydrides, the commonly preferred EPC focus can result in loss of significant geometric, or crystallographic, information, both in real and reciprocal spaces. We posit that such three-dimensional information from electronic band structures, phonon dispersions, and Fermi surfaces may provide key additional insight that complements our understanding of superconducting behaviour. The interpretation of these features and their relative importance for particular compounds are best obtained via appropriately structured DFT calculations.

For example, valuable insights on materials including superconductors have also been gained from questioning the limitations of DFT [35–41]. DFT procedures normally make use of linear approximations [3–5], even though linearity may not properly represent physical realities, particularly for superconductors. Well-constructed discussion papers on the role of anharmonicity in superconductivity have been published and we refer the reader to key articles for further details [39–41]. A further misconception with DFT procedures is that calculated electronic band structures and phonon dispersions cannot provide detailed information on the superconducting gap. It is this specific question that we address in this paper for electronic band structures, as a complement to our earlier recognition that the relative energy of the Kohn anomaly in $MgB_2$ is a good indicator of $T_c$ [42,43].

We have demonstrated, along with others [35–38], that the accuracy of DFT results depends on calculation setup choices, such as the density of k-grid values, cut off energies or radii, pseudopotentials, symmetry assumptions, convergence tolerances, and other resolution-related factors specific to a compound's structure. Variability of results is to be expected for discrete DFT calculations given the variety of possible physical assumptions and setup conditions for calculations, in much the same way as a calculation on integral numbers is approximated by vertical rectangular areas of different widths. Such variability may not significantly affect calculated values in the electron Volt (eV) range. However, for electronic phenomena defined by, for example, the superconducting gap, typically recognised as in the meV range, the degree of variability should be less than the physical value of interest for the calculation to be suited to useful physical interpretation [35,36,38,44].

From a fundamental standpoint, DFT calculations can be conducted, for instance, using algorithms that describe either plane waves or linear combinations of atomic orbitals (LCAO) [45–47]. The plane wave method considers the symmetries of a crystal (as a whole). This approach is suitable for delocalised Bloch-type wave functions, but ignores the symmetry environment of constituent atoms, resulting in loss of detail about the relationship(s) between atomic and crystalline states [48]. Conversely, the LCAO method utilises atomic orbital character, local point symmetry, and directional orbital overlap to define electronic structure and phonon behaviour [45–47]. Thus, the LCAO method explicitly retains the relationship between atomic orbitals and the band structure [47,48] Ultimately, atomic orbital interactions determine the conditions for bonding in crystals, for which orbital directions must overlap, orbital energies must match approximately, and orbitals must have the correct symmetry [49]. This information is more relatable for the experimental chemist who aims at designing and synthesizing new superconducting materials with predicted properties [46,47].

We have shown evidence of superlattice symmetries in $MgB_2$ materials by experiment, from peaks in Raman spectra [44] and, more recently, using synchrotron THz spectroscopy [50]. These THz results display clear increases in spectral density of the low wavenumber absorption peaks when $MgB_2$ samples are cooled below the superconducting transition temperature(s) [50]. These results reveal that $MgB_2$ type materials (i.e., including substituted forms) show lower symmetries than the typically attributed P6/mmm symmetry (space group 191) used for many published investigations. Similar observations of superlattice peaks have been made on a wider range of superconductor materials, including $Nb_3Sn$, $FeSe$, and $YBa_2Cu_3O_{7-x}$ superconductors.

In this work, we further discuss the implications of these experimental results, focused on $MgB_2$, and revisit interpretations of DFT band structure calculations considering superlattice symmetry. As shown below, connections between orbital character and band structure can also be inferred from bands calculated using plane wave methods. Important phase relationships can be identified by the variation in band structure(s) between the reciprocal space centre point, Γ, and positions at the Brillouin zone boundaries. These relationships provide numerical detail on the superconducting gap value and key mechanistic information on superconductivity.

## 2. Results

In our search for the potential origins of superlattices, we revisited the information contained in the DFT calculated electronic band structures reported in earlier publications [44,50]. We have identified additional details on phase relationships embedded in the variation of electronic band energies distributed between the Γ-point and the Brillouin zone boundaries, as described by Stohr and Siegmann [51], Canadell, Doublet and Iung [47], and Sutton [46]. Shifts in band energies from bonding to antibonding character (and vice versa) are identified along different reciprocal directions as discussed below.

### 2.1. Electronic Band Structures: Unit Cells and Superlattices

Figure 1 shows the DFT calculated electronic band structure for $MgB_2$ attributed with typical P6/mmm symmetry. An approximately cosine band shape is apparent for the σ-band along the ΓA direction, with an inflection at the mid-point (ΓA/2) [38,52–56]. A change from bonding to antibonding character is also shown along the ΓA (or $k_z$) direction. In contrast, the σ-bands along the ΓK and ΓM directions show antibonding character at Γ, and then show bonding character at K and M, respectively.

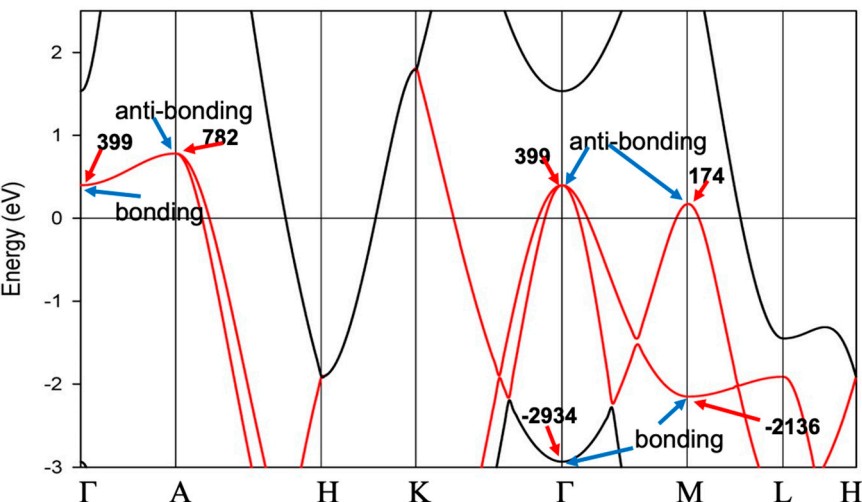

**Figure 1.** Calculated DFT electronic band structure [50] for $MgB_2$ with P6/mmm symmetry (space group 191) and k = 0.005 Å$^{-1}$. The cosine character of the σ-band in the ΓA direction is apparent. Changes from bonding to antibonding character, and vice versa, along different reciprocal directions are labelled. Note the "apparent" inconsistent character for σ-orbital(s) at Γ using P6/mmm, they are both bonding and antibonding. This either means the specific reciprocal direction is important—that is, Γ–A, K–Γ, and Γ–M—or the 2D representation is inappropriate.

These observations are documented in our previous publication [50] and schematics of the bonding and antibonding character at the Γ-point of the σ-bands in respective reciprocal directions are shown in Figures 3 and 4 of reference [50]. The apparent inconsistency in this 2D representation is highlighted in Figure 1 which shows both bonding and antibonding orbitals at Γ suggesting that either the calculation is inaccurate, or the symmetry condi-

tion is inappropriate. Given our comments regarding resolution requirements for DFT calculations [35,36], the latter suggestion is readily evaluated based on our earlier work with superlattice symmetric sub-sets for AlB$_2$-type structures [44]. We present a schematic interpretation of orbital character for conventional and superlattice constructs of MgB$_2$ in Figure 2.

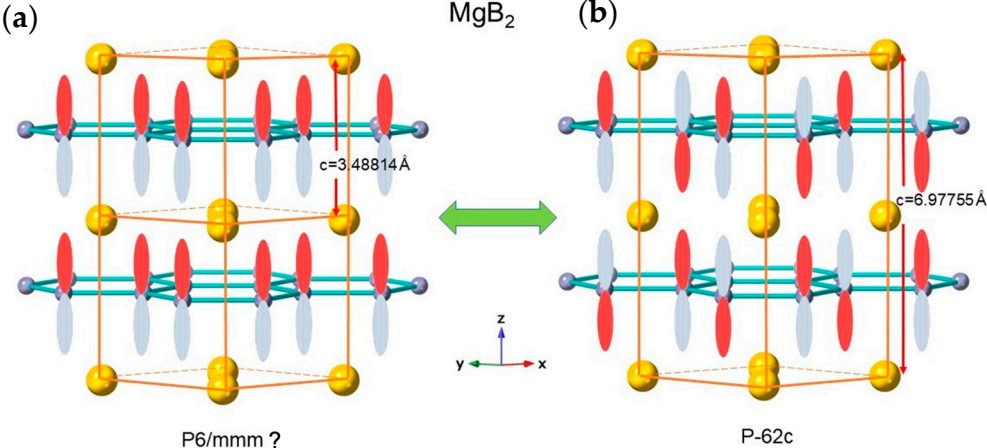

**Figure 2.** (**a**) Attempt at identifying the orbital character at the $\Gamma$ point along the $k_z$ direction, assuming symmetry elements of the P6/mmm group symmetry. Orbital symmetries inferred from band structure results cannot be correctly accommodated. (**b**) Identified bonding character at the $\Gamma$-point of p$_z$ (s)-bands. These schematics are discussed in further detail in reference [50].

### 2.2. Energy Values and Fermi Surfaces

Figure 3 shows an enlarged view close to the Fermi level of the calculated electronic band structure for MgB$_2$ (i.e., <0.9 eV) along the $\Gamma$A or $k_z$ direction, assuming P-6c2 symmetry. This band is a folded representation of the electronic band structure along the $k_z$ direction shown in Figure 1 for P6/mmm symmetry. Energy values at key symmetry points are labelled. These energy values are determined by revised tight-binding equations as described in Section 3.2 and are also displayed on the right of Figure 3. Notice that the bonding and antibonding bands are not completely symmetric, and that the offset from the symmetric average position at $\Gamma$ corresponds to the superconducting gap energy ($\Delta$ = 7.89 meV, 2$\Delta$ = 15.77 meV). Further details on the effect of band folding along the $\Gamma$A direction due to lower symmetry conditions are provided in Supplementary Materials (Figure S2) and as applied to phonon dispersions in reference [50].

We also explore relevant Fermi surface(s) aligned along the *c*-axis using these same DFT calculations for different space group symmetries. In the same fashion that a double superlattice folds the reciprocal directions in a band structure representation, Fermi surfaces also undergo folding. Figure 4a shows the Fermi surface for MgB$_2$ with a P6/mmm symmetry. As noted earlier, the reciprocal space projection for MgB$_2$ normal to *a*\*–*c*\* shows an "hourglass" format of Fermi surfaces, representing the light and heavy effective masses [42]. With DFT calculations based on a lower P-6c2 symmetry, Fermi surfaces for the *a*\*–*c*\* projection—at the same scale—show more clearly the folded tubular sections for the light (Figure 4b) and heavy (Figure 4c) effective masses. Note that similar tubular shapes, albeit with different dimensions along *a*\*, are evident in Figure 4b,c.

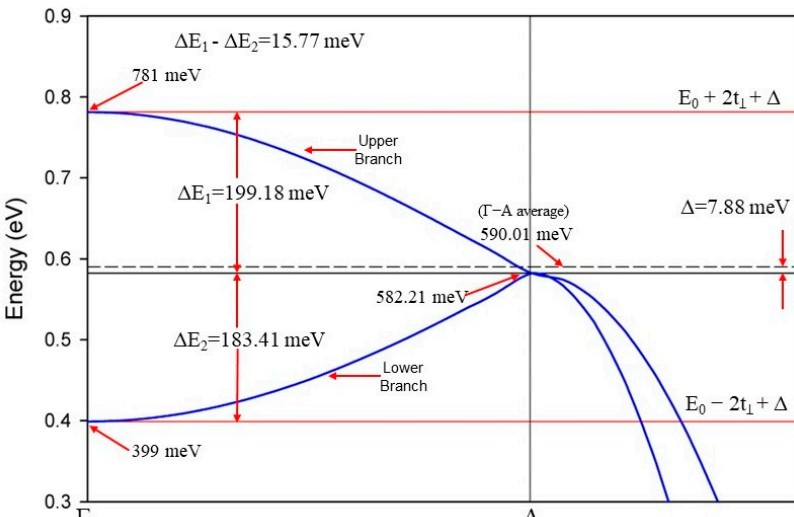

**Figure 3.** Enlarged view of Figure 1 along the ΓA direction for the calculated electronic band (in blue) of MgB$_2$ containing a double superlattice along the *c*-axis with P-6c2 symmetry (space group 190). Energy values in meV at key symmetry points are labelled. Labels to the right are related to the revised tight-binding equations, E$_0$ = 582.21 meV, 2t$_\perp$ = 191.18 meV, D = 7.88 meV (see Section 3.2).

**(a)**

**(b)**

**(c)**

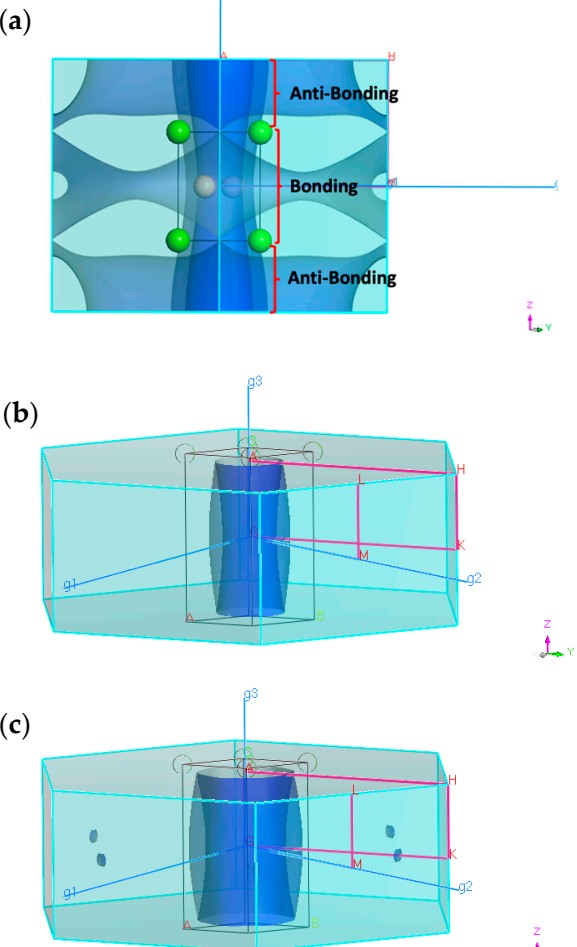

**Figure 4.** Fermi surfaces of MgB$_2$: (**a**) with P6/mmm symmetry, (**b,c**) with P-62c symmetry. (**b,c**) show the folded inner (light effective mass) and outer (heavy effective mass) tubes, respectively.

## 3. Discussion

Through systematic calculations on selected superconductors, we have provided clear evidence that high k-grid resolution is required to reduce, or minimise, the variability of calculated parameters below, or at meV resolution [35,36]. By the same token, high k-grid resolution is required to discern details in calculated electronic band structures and with Kohn anomalies in phonon dispersions that correlate well with the superconducting gap [42,43,57]. This fine k-grid requirement can place a high computational demand on resources, well above the average level generally available and often reported in the published literature [35,36]. For DFT calculations on $MgB_2$ as described below in Methods, we estimate the resolution of calculated energy parameters to within $\pm 1$ meV. Further details on this estimate are provided in the Supplementary section (Table S1).

The presence of superlattice structures in $MgB_2$ has been determined experimentally using Raman and synchrotron THz spectroscopy [44,50], as well as by neutron diffraction [58] and synchrotron X-ray measurements [59]. In actual physical superlattices of scandium-doped $MgB_2$, Kohn anomalies have been proposed as important ingredients for their superconductivity [60]. They have also been proposed as universal ingredients in multigap, multiband pairing mechanisms [61]. More recently, based on quantitative first-principles theory results, Kohn anomalies have been discussed as relevant in a wider range of high $T_c$ superlattice materials [62]. The following discussion focuses on implications for the presence of electron orbital superlattice structures in superconductors using $MgB_2$ as an archetypical example.

### 3.1. Fermi Surface Nesting: Density Waves

Figure 5 shows a schematic for Fermi surface folding of the light effective mass tubular section, based on DFT results in Figure 4. Fermi surface nesting, identified as large sections of a Fermi surface connected by the same vector, is noted in earlier work using P6/mmm symmetry [42,43]. However, nesting is much more obvious and clearly indicated for $MgB_2$ using DFT calculations based on a $2c$ superlattice symmetry such as P-6c2. Similarly, the heavy effective mass tubular section for $MgB_2$ (Figure 4c) shows nesting with slightly longer vectors ($\Gamma M/2$).

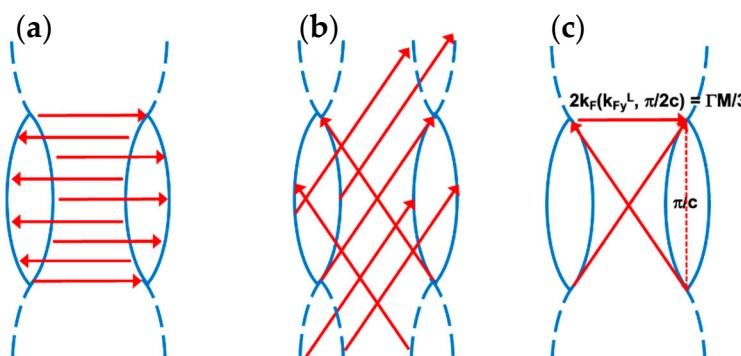

**Figure 5.** Schematics of the folded inner tube Fermi surface for the double ($2\times$) superlattice, for which a DFT calculated Fermi surface is shown in Figure 4. Fermi surface nesting is prevalent for the double superlattice with a vector approximately: (**a**) $\Gamma M/3$ long along the horizontal direction; (**b**) $\Gamma M/3$ + p/c z along the diagonal direction. The horizontal projection of the diagonal nesting vector is $\Gamma M/3$, as shown in (**c**).

Nesting also occurs between the same sides of light and heavy effective mass tubular sections (with horizontal vector $\Gamma M/12$), as shown in supplementary Figure S2. The magnitude of this short horizontal nesting vector is equal to the difference in light and heavy effective mass Fermi wavevectors. Each tubular section gives rise to three sets of nesting vectors: one horizontal and two in diagonal directions. The Fermi wavevector at the inflection point is half the magnitude of the longest horizontal nesting vector. Therefore, the nesting vectors along the $\Gamma M$ direction are approximately $\Gamma M/3$ and $\Gamma M/2$ for the

light and heavy effective mass tubular sections, respectively (see Supplemental Figure S9 in reference [50] and Figure S1). Diagonal nesting vectors include the same horizontal component as the horizontal nesting vector and $\pi/c$ as a perpendicular component.

Most, if not all, nesting vectors connect a bonding to an antibonding section of the Fermi surface, including for the horizontal short nesting vector (as shown in Figure 5). At a certain instant of time, we expect that the bonding orbital is populated with electrons, while the antibonding orbital is empty. If a nesting vector is associated with electrons transferring or hopping synchronously from bonding to antibonding orbitals, then we also expect that the orbital population will be periodically shifted or displaced consistently with the symmetry of the site. In this way, we can recognise that oscillating bonding and antibonding character with nesting vectors connecting these two orbital elements suggest a resonating behaviour defined by the structural symmetry of $MgB_2$.

The theory for density waves in solids indicates that electron–hole pairs of states, for which the nesting condition applies, develop modulations with periodicity determined by the nesting vector $\mathbf{Q} = (2\mathbf{k}_F, \pi/c)$ [63]. Such a nesting vector, when the same vector connects substantial sections of the Fermi surface, is perfect or nearly perfect. This type of nesting vector contributes a divergence or singularity to the Lindhard response function. This Lindhard function relates an induced charge to the action of an external time independent potential.

A charge density wave has a modulation with wavevector $\mathbf{q}_{\parallel} = 2\mathbf{k}_F$ in the in-plane direction. In the perpendicular direction, the wavevector, $\mathbf{q}_{\perp} = \pi/c$, signals that planes separated by $c$ are out of phase. The period of a charge density wave is related to $\mathbf{k}_F$ by $\lambda_o = \pi/\mathbf{k}_F$. For a small dispersion in the $k_z$ direction, proximity to the Fermi energy and nearly perfect nesting, the following equations apply [63]:

$$E(\boldsymbol{k}) = E_0 - 2t_c\cos(k_z c), \tag{1}$$

$$k_{x,y}(k_z) = k_F + \left(\frac{2t_c}{v_F}\right)\cdot\cos(k_z c). \tag{2}$$

These equations, originating in principle from the perspective of Fermi surface nesting, resemble tight-binding equations used to describe the tubular sections of the Fermi surface around $\Gamma$A for $MgB_2$ [38,52–56]. Thus, nesting conditions that normally lead to charge density waves are ubiquitous within the Fermi surface topology of an $MgB_2$ double superlattice along the $c$-axis. We believe this condition may apply more generally to the Fermi surfaces of many other layered superconductors, if not all layered forms, since these are all approximately described by identical types of tight-binding equations [52–56].

### 3.2. Revised Tight-Binding Equations

With accurately determined energy values at symmetry points based on electronic band structure calculations, we revisit tight-binding equations for the bonding and antibonding branches of the σ-bands in the $k_z$ direction. For this analysis, accuracy is important because small deviations in the order of several meV's could be sufficient to blur and mask the clear and systematic identification of gap energy from the electronic band energy.

Tight-binding equations are normally written as [50]:

$$E_n\left(k_x, k_y, k_z\right) = E_0 - 2t_{\perp}\cos\left(ck_z\right) - \hbar^2\left(k_x^2 + k_y^2\right)/2m_n,\, 0 \leq ck_z \leq \pi \tag{3}$$

where $t_{\perp}$ is the transfer coefficient and $m_n$ are the heavy and light effective masses.

However, careful inspection of the energy values at key symmetry points indicates that these energies are not part of the same, single cosine function. The energies in the antibonding branch of the Fermi surface are slightly offset by a value $2\Delta$ of energy (for a

given $k_z$) from the energies of the bonding branch of the Fermi surface. Hence, we adjust Equation (3) as follows:

$$E_n\left(k_x, k_y, k_z\right) = E_0 - \left(2t_\perp^n - \Delta\right)\cos(ck_z) - \hbar^2\left(k_x^2 + k_y^2\right)/2m_n,$$
$$0 \leq ck_z \leq \frac{\pi}{2} \tag{4}$$

$$E_n\left(k_x, k_y, k_z\right) = E_0 - \left(2t_\perp^n + \Delta\right)\cos(ck_z) - \hbar^2\left(k_x^2 + k_y^2\right)/2m_n,$$
$$\frac{\pi}{2} \leq ck_z \leq \pi \tag{5}$$

where we add the index $n$ for heavy and light effective masses to the transfer coefficient, because under pressure the curvatures of the surfaces change.

The bonding and antibonding energy bands in the $k_z$ direction (as shown in Figure 3) show a strong similarity to the extended Hückel approach for orbitals and bands of a chain of (dimerised) hydrogens (see for example, Figure 3.19 of reference [47]). In this approach, for the same $k_z$, two degenerate Bloch orbital branches interact, as a simple addition or subtraction, to produce the orbitals of a solid [47]. The revised tight-binding equations for MgB$_2$ derived above also indicate that a phonon is lost (or gained) in the bonding branch when an electron transfers or hops into (or from) the antibonding branch (see Supplementary Equations (S1)–(S3)). These equations also show that the process involves out-of-phase electrons travelling in two opposite directions ($k_z$ and $-k_z$), implying that wave reflection(s) may occur. An electron can only be scattered if a phonon is emitted or absorbed and if the total wave vector is not changed, except by a reciprocal lattice vector [64,65]. On the other hand, the resulting ions cannot follow the electronic motion and, thus, these ions experience a time-averaged adiabatic electronic potential [66].

In order to include a proper sinusoidal wave traveling in the $k_z$ direction at frequency $\omega$, a factor incorporating time, $t$, is required in Supplementary Equations (S1)–(S3). In our previous publications and recognised by others [67,68], we show that the electronic band structure is a frozen picture of the equilibrium position. However, when vibrations are included the dynamic electronic band structure displays degeneracy breaking and oscillatory swings of the $\sigma$-bands across the Fermi energy. For MgB$_2$, this is particularly apparent for $E_{2g}$ or acoustic modes [67–69]. These swings are accompanied by dynamic electron density (wave) distributions that screen the $\sigma$-bonding regions with increased ion core repulsions from approaching boron atoms.

Screening is achieved by depleting electrons from the $\sigma$-bonding regions with reduced ion core repulsions from more distant boron atoms. Such electron density waves essentially behave as dynamic, periodic dimerisations (or n-merisations) [47]. As already mentioned, the bonding–antibonding character of the MgB$_2$ sp$^2$ orbitals lends susceptibility to resonant behaviour. This resonant behaviour—moderated by the superlattice symmetry—remains in operation for instances when the sign(s) of sp$^2$ orbitals invert (i.e., if bonding and antibonding orbital lobes change sign). Thus, a resonance between equivalent bonding configurations is induced below the superconducting transition temperature in a similar fashion to the resonance of benzene bonds [49,70].

### 3.3. Identification of Cooper Pairing

Figure 6 shows additional potential geometric relationships for electrons moving along electronic bands or nested Fermi surfaces.

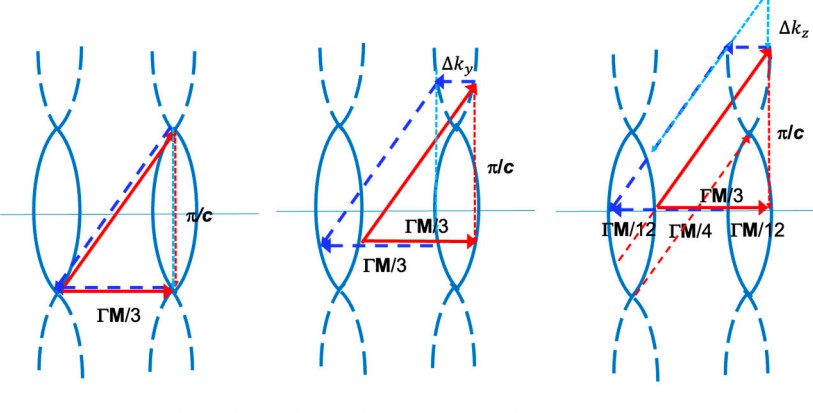

$$(\Delta k_y, \Delta k_z): \quad (0,0) \rightarrow (\Gamma M/12, \pi/4c) = \frac{1}{4}\,(\Gamma M/3, \Gamma A)$$

**Figure 6.** Schematics of additional geometric relationships and nesting vectors between sections of Fermi surfaces which could be associated with interactions leading to Cooper pairing. See text for more details.

Referring to Figure 6, we consider that the Fermi vectors connected by diagonal nesting vectors, $\mathbf{q}_\Delta$, undergo scattering. Then, Equations (4)–(6), describe the scattering processes.

$$\mathbf{k_F}^1 + \mathbf{q}_\Delta = \mathbf{k_F}^{1'}, \tag{6}$$

$$\mathbf{k_F}^2 - \mathbf{q}_\Delta = \mathbf{k_F}^{2'}, \tag{7}$$

$$\mathbf{k_F}^1 + \mathbf{k_F}^2 = \mathbf{k_F}^{1'} + \mathbf{k_F}^{2'} \tag{8}$$

$$\mathbf{q}_\Delta = \Gamma A + \Gamma M/3 \tag{9}$$

These equations are of a form that can be identified as Cooper pairing [64,65,71]. Indeed, energy is also conserved since $E_F' = E_F \pm \Delta = E_F \pm 2\hbar\omega$ (see Figure 3 and Supplementary Information). From geometric relationships, we can also determine the following:

$$\frac{\frac{5\Gamma M}{12}}{\frac{\Gamma M}{3}} = \frac{\frac{\pi}{c} + \Delta k_z}{\frac{\pi}{c}} \tag{10}$$

$$\frac{5}{4} = 1 + \frac{c\Delta k_z}{\pi} \tag{11}$$

$$\Delta k_z = \frac{\pi}{4c} \tag{12}$$

Therefore, a quadruple (4×) periodicity detected in THz and neutron scattering experiments [58] reflects prima facie, in an uncomplicated manner, geometric relationships that favour enhanced nesting and coherency.

## 4. Methods

DFT calculations were undertaken using the CASTEP module of Materials Studio 2018 and more recent upgrades of this software. Both the linear response within the local density approximation (LDA) and generalized gradient approximations (GGA) with a dense k-grid mesh (k < 0.005 Å$^{-1}$) are used as indicators of the spread of results for specific conditions, as detailed in our earlier investigations [42,43,57,67]. Calculations are undertaken with an ultra-fine cut-off (typically 990 eV). Convergence criteria for most calculations are as follows: energy at $5 \times 10^{-6}$ eV per atom; maximum force at 0.01 eV; maximum stress at

0.02 GPa; and maximum atom displacement at $5 \times 10^{-4}$ Å. Optimal calculation conditions using CASTEP for the $AlB_2$-type structure are given in earlier works [35,36,42].

## 5. Conclusions

We show that electronic bands perpendicular to the layers of superconductors with a layered structure can be better approximated by revised tight-binding equations. Applying folding of the reciprocal space unit cell, motivated by prior experimental measurements that show evidence of superlattices, reveals a very extensive amount of previously unreported nesting of the superlattice folded Fermi surface. The phenomenology arising from nesting, such as the generation of charge, or spin, density waves, are expected and provide further detail on the cosine dependence of tight-binding bands in superconductors. Nearly parallel Fermi surfaces, associated with inverted parabolic bands (representing light and heavy effective mass(es) [35,36,38]) are key to the unique Fermi surface topology of superconductors. Such topology has already been shown to give rise to Kohn anomalies that correlate exceptionally well to the transition temperature for various compositions of superconductors and with pressure [42,43,57,72,73].

The point $\Gamma A/2$ of the original P6/mmm cell (which is equivalent to the reciprocal point A of the double supercell along the c-axis for $MgB_2$) locates a singularity where the character of the bands changes from bonding to antibonding. Electrons approaching this nodal point must have either of these characters while, at the nodal point, the electron character is undefined or singular. The DFT methodology incrementally shifts electrons around the electronic band structure to minimise the energy of a compound given a specific structural symmetry. However, movement of electrons while the structure is approaching a converged configuration involves emission or absorption of phonons. This is consistent with electron movement involving a scattering process, and as mentioned, an electron can only be scattered if a phonon is emitted or absorbed [64,65]. With high-resolution DFT calculations using an appropriate k-grid, fine granular detail of electronic band structures can be discerned for superconductors. Our preliminary inspection of electronic band structures for other layered superconductors suggests that similar symmetry-derived corrections to tight-binding equations may enable direct determination of superconducting band gap(s).

Revised tight-binding equations based on high-resolution band structure calculations reveals additional terms are essential in order to accommodate the superconducting gap and to appropriately balance band energies at key symmetry points. For $AlB_2$-type structures, bonding–antibonding relationships and the cosine function of tight-binding equations suggest the presence of out-of-phase parallel planes. Waves propagating in opposite directions, with opposite phase to the primary cosine term, are also consistent with this analysis and include emission or absorption of phonons. Furthermore, Cooper pairing linked to the folded Fermi surfaces and involving the nesting vector can be readily identified, providing unprecedented insight on mechanisms for superconductivity. Due to computational limits related to DFT methodology, we are unable to demonstrate the time dependence of these phenomena as appropriately recognised by the time-dependent Schrödinger equation. We encourage practitioners of time-dependent DFT (TD-DFT) to revisit these results for a potential, more inclusive, and self-consistent time-dependent approach.

Evidence of superlattices in vibrational spectra from a wide variety of superconductors and preliminary extensions of our calculations on $MgB_2$ at higher pressures, to FeSe with pressure, and to other compounds suggest that the methodology presented here has general applicability to other superconductors. Establishing strong connections between electronic band structures and Fermi surfaces to atomic orbital characteristics may be a practical complementary tool for the experimental chemist to use to design and synthesize new superconductors.

**Supplementary Materials:** The following supporting information can be downloaded at: https://www.mdpi.com/article/10.3390/condmat8030072/s1, Table S1. Summary of k-grid dependence for band energies at key symmetry points for $MgB_2$ with P6/mmm or P-6c2 symmetry. Standard deviations for P-6c2 calculations for $k < 0.013$ A$^{-1}$ are less than 1 meV; Figure S1. Fermi surfaces for $MgB_2$ with attributed P6/mmm symmetry. Approximate geometrical relationships for the radii of the light (inner) and heavy (outer) effective mass tubular sections are labelled (see also Figure S9 of reference [50] in the article).

**Author Contributions:** Conceptualization, J.A.A. and I.D.R.M.; methodology, J.A.A. and I.D.R.M.; software, J.A.A.; validation, J.A.A. and I.D.R.M.; formal analysis, J.A.A.; investigation, J.A.A. and I.D.R.M.; data curation, J.A.A. and I.D.R.M.; writing—original draft preparation, J.A.A.; writing—review and editing, I.D.R.M.; visualization, J.A.A. and I.D.R.M. All authors have read and agreed to the published version of the manuscript.

**Funding:** This research received no external funding.

**Data Availability Statement:** The data presented in this study are available in Supplemental Information and in [50].

**Acknowledgments:** The authors are grateful to the Central Analytical Research Facility (CARF) and the e-Research Office, both at QUT, for access to laboratory facilities and to high-performance computing and assistance, respectively. While this article has not directly used previous results at the THz beamline of the Australian Synchrotron, the authors have benefited greatly from inspiration motivated by the prior published results cited above.

**Conflicts of Interest:** The authors declare no conflict of interest.

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
