# Peer review of "Superlattices, Bonding-Antibonding, Fermi Surface Nesting, and Superconductivity"

_condensedmatter, doi:10.3390/condmat8030072_

Round 1
Reviewer 1 Report
The manuscript by Alarco et al. reports a very interesting work on the properties and superlattice symmetries of MgB2 and related materials-systems. The work has a focus on the energy band theory and Fermi surface calculation, revealing presence of Fermi surface nesting and topological effects. The presented results are suitable for publication in the Journal (Condensed Matter). I recommend acceptance in the present form.
Author Response
Thank you for the positive feedback.
Reviewer 2 Report
Previous observations of THz spectra show superlattice absorption peaks with low wavenumber, for which spectral density evolves and intensifies after cooling below the superconducting transition temperature for MgB2. This indicates connections between electron orbital superlattices and superconducting mechanisms. This work reproduce the orbital superlattice features in MgB2 using DFT and other simulation methods. Systematic studies of these simulations enable authors to conduct DFT re-interpretation of BCS superconductivity for MgB2.
This is a great theoretical work on the superconductivity mechanism of MgB2. It provides new perspective of understanding conventional BCS theory connected with long range structural orders. I recommend to publish this manuscript in present form.
Author Response
Thank you very much for the very positive comments on our work.
Reviewer 3 Report
Since 2001, MgB2 has been identified as a superconductor with a transition temperature of 39K. Over the past two decades, significant attention has been directed towards studying MgB2. The superconductivity and mechanism of MgB2 have been largely elucidated, revealing a conventional BCS superconducting mechanism. This article employs first principles and Tight binding approaches to investigate the superconducting properties of MgB2, further substantiating its BCS-type superconducting mechanism. I recommend its publication; however, I suggest that the authors make detailed revisions to both the abstract and the main text.
Upon reviewing the abstract, it seems that the authors have conducted both experimental and theoretical research. However, upon closer examination, it becomes apparent that the study exclusively focuses on the theoretical aspects of MgB2. To avoid any confusion, it is crucial to accurately convey the nature of the investigation.
Additionally, given the abundance of existing experimental and theoretical research on MgB2 and the already well-established understanding of its superconductivity, the authors should emphasize and highlight the innovative aspects of this particular article. Emphasizing the novel contributions will help readers understand the distinctiveness of the research in the context of the broader body of knowledge on MgB2 superconductivity.
Moderate editing of English language required
Author Response
We are very grateful for the constructive comments of the reviewers.
In the abstract we have emphasized that the experimental results are from earlier work. We also point out more clearly what is done in this work.
In the introduction, we have added two sentences that identify more clearly some of the objectives of this work:
- The interpretation of these features and their relative importance for particular compounds are best obtained via appropriately structured DFT calculations.
- It is this specific question that we address in this paper for electronic band structures as a complement to our earlier recognition that the relative energy of the Kohn anomaly in MgB2 is a good indicator of Tc (54, 55).
In the discussion, we have added a few sentences with references that further link superlattices with Kohn anomalies:
In actual physical superlattices of scandium-doped MgB2, Kohn anomalies have been proposed as important ingredients for their superconductivity (59). They have also been proposed as universal ingredients in multigap, multiband pairing mechanisms (60). More recently, based on quantitative first-principles theory results, Kohn anomalies have been discussed as relevant in a wider range of high Tc superlattice materials (61).
In the conclusions, we have added two sentences, one to re-emphasize that experimental results are from prior work and to highlight the novelty of this article, and another to bring awareness about TD-DFT.
- Applying folding of the reciprocal space unit cell, motivated by prior experimental measurements that show evidence of superlattices, reveals a very extensive amount of previously unreported nesting of the superlattice folded Fermi surface.
- 2. We encourage practitioners of time-dependent DFT (TD-DFT) to revisit these results for a potential, more inclusive and self-consistent time dependent approach.
Reviewer 4 Report
The manuscript provides a technical and physically insightful description of how density-functional theory (DFT) calculations and model tight-binding calculations can be used to investigate phenomena leading to superconducting transitions in MgB2 and, potentially, other layered materials. Rather than presenting new calculations, the authors reassess results from their earlier work (specifically Refs. 41 and 47), with the aim of explaining the potential origin of the electron orbital superlattices experimentally observed using Raman and THz absorption spectroscopy in MgB2. They show, in particular, that the orbital character (bonding/anti-bonding) of bands approaching specific nodal points in reciprocal space progressively become undefined and that this process must be accompanied by electron scattering in the form of phonon absorption or emission. Further insight into this process can be gained using tight-binding equations based on band structures calculated at high k-point resolution. The work provides evidence for the existence of a link between electronic orbital character, its dependence on the system's symmetry and the nature of the band structure and Fermi surface topology in layered material.
The work is clear, focussed, and well described. The level of detail is appropriate. The main conclusions are well supported by the material discussed. I think the work is original and makes an important contribution to understanding the origin of superconductive behavior in MgB2 and related materials. I recommend publication, but I would like the authors also to address the minor points listed below.
1) The authors indicate that, in the DFT calculations, a high-quality k-point grid for the representation of the band structures is a crucial requirement. On the other hand, there are other approximations in standard DFT implementations that (one would think) could play a role. For instance, the authors mention that the DFT calculations were carried out at the LDA or GGA levels. Although for the specific case of MgB2 these functionals may have given good accuracy, would quantitative estimates of Fermi surface topology and superconducting properties be impacted by the approximations of LDA/GGA? Would more advanced approaches for the calculation of quasi-particle energies (e.g., Green's function based methods) provide more robust estimates of band structure properties for MgB2-like layered materials, or is LDA/GGA DFT, in general, the most appropriate choice?
2) In the Conclusions the authors mention that Cooper pair formation linked to folded Fermi surfaces and the time-dependent phenomena related to it cannot be addressed using DFT. Although this is correct, there are extension of DFT that can account for the dynamical response of the electrons, like time-dependent DFT (TD-DFT). To the authors' knowledge, have there been any attempts to model these phenomena using TD-DFT, or are there further complication that make this approach impractical.
3) In figure 3, the first label along the x axis should be Gamma, rather that G, for consistency with the other Figures.
Author Response
With respect to whether LDA, GGA or alternative approaches would be more accurate, we cannot really add more informed comments. LDA/GGA have worked rather well so far for the investigated compounds and provide a spread of results, as stated already.
Both the linear response within the local density approximation (LDA) and generalized gradient approximations (GGA) with a dense k-grid mesh (k < 0.005Å-1) are used as indicators of the spread of results for specific conditions, as detailed in our earlier investigations (54, 55, 56, 66).
Computational approaches that add complexity and cost would have to be justified. They are outside the scope of this article.
In the conclusions, we have added two sentences, one to re-emphasize that experimental results are from prior work and to highlight the novelty of this article, and another to bring awareness about TD-DFT.
- Applying folding of the reciprocal space unit cell, motivated by prior experimental measurements that show evidence of superlattices, reveals a very extensive amount of previously unreported nesting of the superlattice folded Fermi surface.
- We encourage practitioners of time-dependent DFT (TD-DFT) to revisit these results for a potential, more inclusive and self-consistent time dependent approach.
Reviewer 5 Report
The paper is well written, and is related to very interesting material MgB2.
It is a theoretical explanation of their earlier synchrotron experimental data.
I think it can be published as it is written now. It contains new research
and it is acceptably written.
Language is ok.
Author Response
Thank you for the positive review.
Round 2
Reviewer 3 Report
I recommend its publication.
No Comments